# Proteasome Inhibitor MG-132 and PKC-ι-Specific Inhibitor ICA-1S Degrade Mutant p53 and Induce Apoptosis in Ovarian Cancer Cell Lines

**DOI:** 10.3390/ijms26073035

**Published:** 2025-03-26

**Authors:** Mahfuza Marzan, Nuzhat Nowshin Oishee, Abigail Oluwafisayo Olatunji, Abiral Hasib Shourav, Radwan Ebna Noor, Aaron Joshua Astalos, James W. Leahy, Mildred Acevedo-Duncan

**Affiliations:** Department of Chemistry, University of South Florida, 4202 E Fowler Ave, CHE 205, Tampa, FL 33620, USA; mmarzan@usf.edu (M.M.); nuzhatn@usf.edu (N.N.O.); abigailolatunji@usf.edu (A.O.O.); hasibshourav@usf.edu (A.H.S.); noor2@usf.edu (R.E.N.); aaronastalos@usf.edu (A.J.A.); jwleahy@usf.edu (J.W.L.)

**Keywords:** ovarian cancer, p53, PKC-ι, proteasome inhibitor MG-132

## Abstract

Ovarian cancer is the most lethal gynecological cancer, with a 5-year survival rate of approximately 50%. Mutation in the p53 gene and overexpression of the atypical protein kinase C iota (PKC-ι) are two phenomena widely manifested in ovarian cancer. This study investigated the role of PKC-ι-specific inhibitor ICA-1S and proteasome inhibitor MG-132 in ovarian cancer cell lines. To discern the result, cell proliferation assays, cytotoxicity assays, Western blotting, immunofluorescence, flow cytometry, small interfering RNA, and co-immunoprecipitation techniques were applied. ICA-1S and MG-132 were found to inhibit the proliferation of ovarian cancer cell lines significantly. ICA-1S reduced the level of oncogenic PKC-ι as expected. In addition, ICA-1S and MG-132 both were able to decrease the level of mutated p53 in the ES-2 cell line through separate pathways. On the contrary, MG-132 increased the level of wild-type p53 in the HEY-T30 cell line by inhibiting proteasomal degradation. MG-132 also induced apoptosis and autophagy in the ovarian cancer cell lines. We concluded that ICA-1S alone or in combination with MG-132 could be a potential treatment for mutated p53-containing and PKC-ι-overexpressing ovarian cancers.

## 1. Introduction

Ovarian cancer is the most lethal gynecological cancer in today’s world, and its lethality is mostly attributed to the late stage of diagnosis, resistance against chemotherapeutics such as platinum drugs or taxanes, genetic mutations in tumor suppressors, and hormonal and environmental factors [1,2,3,4]. In the year 2020, there were 313,959 new cases of ovarian cancer, and 207,252 women died of this disease worldwide [5]. Of all types, epithelial ovarian cancer is the most common type of ovarian cancer, of which high-grade serous ovarian cancer (HGSOC) is the most aggressive and presents a poor prognosis with a 5-year survival rate of around 25%. Approximately 94–96% of HGSOC cases are characterized by the presence of a mutation in the p53 protein (Mutp53) with loss of function of wild-type p53 (WTp53) or gain of functions of Mutp53 leading to tumorigenesis [6,7,8,9]. The tumor suppressor p53 has its role as a classical transcription factor in transcribing proteins that play a role in apoptosis, senescence, metabolism, autophagy, invasion, metastasis, and regulation of the tumor microenvironment and are regulated tightly in normal cells [10]. Whereas the degradation of WTp53 is regulated particularly by the MDM2 (murine double minute 2, E3 ligase of p53 protein)-mediated ubiquitin–proteasome pathway (UPP), Mutp53 degradation via UPP can be regulated by diverse proteins [11,12].

The UPP is an important protein degradation and turnover pathway that influences more than 80% of specific protein degradation in the cellular environment and also plays an important role in cellular homeostasis. In cancerous cells, the use of proteasomal degradation-specific inhibitors (proteasome inhibitors, PIs) was found to induce apoptosis [13]. In ovarian cancer cell lines PIs, especially MG-132 (carbobenzoxy-L-leucyl-L-leucyl-L-leucinal), induced apoptosis and autophagy and reduced cell proliferation when combined with platinum drugs [14,15]. Interestingly, one study reported the involvement of MG-132 and some other PIs in partially suppressing mutant p53 protein through the upregulation or stabilization of MDM2 [16]. However, the molecular mechanism behind the downregulation of Mutp53 by MG-132 has not yet been fully explored.

Recently, atypical protein kinase C iota (PKC-ι), a serine/threonine kinase, has been detected as an oncogene in many cancers such as lung cancer, triple-negative breast cancer, and some other cancers. It has been detected as one of the key players in ovarian cancer pathogenesis, playing a role in glycolysis, proliferation, and ovarian serous tumor initiation and growth [17,18,19,20]. Often PKC-ι is found to be overexpressed in ovarian cancer tissue samples compared to normal ovarian samples and is correlated with reduced median survival time. PKC-ι, as a kinase, can influence the Kirsten rat sarcoma viral oncogene homolog–mitogen-activated protein kinase (RAS-MAPK) pathway, the phosphatidylinositol 3-kinase and protein kinase B (PI3K-AKT) pathway, and many more. Mutation in KRAS, PI3KA, or BRAF has been well documented in ovarian cancer [3]. However, targeted drugs for these pathway proteins or PKCs have not shown any significant positive outcomes in ovarian cancer clinical trials [21].

In this study, a novel experimental drug, PKC-ι-specific inhibitor ICA-1S (5-amino-1-2,3-dihydroxy-4-(methylcyclopentyl)-1H-imidazole-4-carboxamide), was implemented. ICA-1S has previously shown promising efficacy in neuroblastoma, renal cell carcinoma, and some other cancers by inhibiting PKC-ι and also showed lower toxicity in a preclinical trial [22,23,24]. We report that the inhibition of PKC-ι by ICA-1S significantly reduces the proliferation of ovarian cancer cell lines. It was also found that MG-132 could substantially reduce ovarian cancer cell viability. In addition, we observed that the application of ICA-1S can cause the downregulation of Mutp53 by partially preventing the phosphorylation of p53 by PKC-ι and making it vulnerable to proteasomal degradation. Whereas ICA-1S was also affecting the level of WTp53 to some extent, the addition of MG-132 could reverse this degradation phenomenon and increase the level of WTp53 compared to the control. Interestingly, MG-132 alone was observed to downregulate the Mutp53 and to upregulate the WTp53 possibly by activating the autophagy pathway and by inhibiting proteasomal degradation, respectively. Moreover, the knockdown of PKC-ι by siRNA (small interfering RNA) increased the protein level of WTp53 in the HEY-T30 cell line, emphasizing the oncogenic role of PKC-ι in ovarian cancer. Therefore, ICA-1S alone or in combination with MG-132 can provide a unique option to treat mutated p53-containing and oncogenic PKC-ι-overexpressing ovarian cancer.

## 2. Results

### 2.1. ICA-1S and MG-132 Have an Inhibitory Effect on Ovarian Cancer Cell Proliferation

ICA-1S significantly inhibited the proliferation of all ovarian cell lines (ES-2, HEY-T30, and OVCAR-3) in a dose-dependent manner. The inhibitory concentration that reduced the proliferation to 50% (half maximum inhibitory concentration, IC_50_) was 15 μM, 25 μM, and 45 μM for ES-2, HEY-T30, and OVCAR-3 cell lines, respectively (*p* < 0.001). To determine the effect of MG-132 on the cell viability of ovarian cancer cell lines, we used the water-soluble tetrazolium (WST-1) assay. In HEY-T30 and OVCAR-3 cell lines, MG-132 reduces cell viability at the lowest 0.5 μM concentration at a single dose (*p* < 0.001); however, in the ES-2 cell line, the lowest dose that could significantly affect cell viability was 1.5 μM (*p* < 0.05) (Figure 1).

### 2.2. MG-132 Induces Ovarian Cancer Cell Death

MG-132 induced cell death in ovarian cancer cell lines as indicated by the presence of either cleaved caspase-3 or cleaved PARP (poly-ADP ribose polymerase) or both in the Western blot (Figure 2A). ICA-1S (IC_50_) did not show any significant induction of cellular death, although the combination of ICA-1S (IC_50_) with MG-132 (2 μM) showed the highest level of cleaved caspase 3 and cleaved PARP protein in ES-2 and HEY-T30 cell lines.

Flow cytometry analysis with Annexin-V APC (Allophycocyanin) and propidium iodide (PI) was performed on ES-2 and HEY-T30 cell lines to detect apoptosis. Early apoptosis was found to be induced by ICA-1S, and late apoptosis was induced by MG-132 (Figure 2B).

### 2.3. Level of WT p53 and Mutp53 Are Regulated Differently by MG-132 and ICA-1S

Levels of Mutp53 and WTp53 were shown to be regulated differently by MG-132 and ICA-1S in the ovarian cancer cell lines as seen in the Western blot results. In the ES-2 cell line, which has the mutation in p53, ICA-1S at 15 μM concentration could significantly reduce Mutp53 (45.79%, *p* < 0.001), whereas, in the HEY-T30 cell line, the same dose caused degradation of WTp53 to some extent (64.25%). The level of Mutp53 was significantly downregulated by MG-132 alone (40.20%, *p* < 0.001) or in combination with ICA-1S (19.14%, *p* < 0.001). The level of the phosphorylated form of mutated p53 (Pp53, ser-15) was also downregulated (Figure 3). Immunofluorescence staining of Mutp53 showed the same pattern of downregulation in the ES-2 cell line (Figure 4).

In contrast, in the HEY-T30 cell line that contains WTp53, MG-132 treatment could increase the level of WTp53 compared to control when applied alone (220.62%) or in combination with ICA-1S (157.60%). The level of phospho-p53 also increased when treated with MG-132. However, in the cells treated with ICA-1S alone, there was the degradation of WTp53 in the HEY-T30 cell line indicating the p53 degradation pathway mediated by MG-132 and ICA-1S could be different. In both cell lines, MG-132 could increase the level of MDM2 significantly (*p* < 0.05) (Figure 3). The protein level of MDM2 was not significantly affected by ICA-1S in both cell lines, demonstrating that MDM2 is not solely responsible for the downregulation of p53 in those cell lines.

### 2.4. Autophagy May Play a Role in the Degradation of Mutp53 by MG-132

Autophagy may play a role in the degradation of Mutp53 as the autophagosome seems to be activated by the treatment of MG-132. The level of inhibitory phosphorylation on serine-757 in ULK-1 (Unc-51 like autophagy activating kinase) was decreased in all treated samples except only when treated with a low concentration of MG-132. This phosphorylation is supposed to be performed by mTORC1 (mammalian target of rapamycin complex 1), which hinders the interaction between ULK-1 and AMPK (AMP-activated protein kinase) and inhibits autophagy [25]. The level of LC3 A/B-II (microtubule-associated protein 1A/1B light chain 3 alpha/beta II), an indicator of autophagy occurrence, was increased upon treatment with MG-132 in both cell lines, however, not with the treatment with ICA-1S (Figure 5).

### 2.5. Level of Atypical Kinases and Association of PKC-ι with p53

All treatments decreased the levels of atypical kinases PKC-ι and PKC-ζ (protein kinase C zeta) in the HEY-T30 cell line. In the ES-2 cell line, the level of PKC-ι protein was downregulated; however, PKC-ζ protein level increased with ICA-1S, MG-132, and with the combination of both drugs (Figure 6A).

In the co-immunoprecipitation experiment, where PKC-ι was co-immunoprecipitated with its associated protein, its association with p53 was revealed in the HEY-T30 cell line. Compared to the HEY-T30 cell line, PKC-ι of the ES-2 cell line showed less association (Figure 6C).

### 2.6. siRNA Inhibition of PKC-ι Does Not Affect the Mutp53 but Increases WTp53

Knockdown of PKC-ι using siRNA (small interfering RNA, 27-mer) reduced the expression of PKC-ι protein but increased the level of WTp53 in HEY-T30 significantly (*p* < 0.05). Although the level of Mutp53 was slightly decreased with the knockdown of PKC-ι, it was statistically insignificant (Figure 6B).

## 3. Discussion

Late-stage diagnosis, high relapse rate, and resistance to common therapeutics pose challenges to the success of ovarian cancer treatment [26]. Mutation in p53 as well as overexpression of the PKC-ι oncogene are two biological phenomena identified in ovarian cancer, and several studies have been performed to determine targeted treatments to handle this condition [17]. In this study, we sought to find the effect of the PKC-ι-specific inhibitor ICA-1S and the proteasomal inhibitor MG-132 on ovarian cancer cell lines.

The mode of action of ICA-1S as determined in previous publications using molecular docking is the inhibition of PKC-ι by binding to the 469–475 amino acid residues [27]. In this study, ICA-1S was able to reduce the proliferation of ES-2, HEY-T30, and OVCAR-3 cell lines to 50% (IC_50_) at 15 μM, 25 μM, and 45 μM concentration in vitro (*p* < 0.001). Previously, ICA-1S was reported to have inhibitory effects in different cancer cell lines; however, there are differences in dose–response curves and IC_50_ values in various cancer types [22]. In contrast, in this study MG-132 was found to reduce cell viability of ovarian cancer cell lines in a dose-dependent manner; the highest dose of 2 μM applied once for 18 h showed the most significant reduction in cancerous cell viability. At 2 μM concentration, MG-132 reduced the viability of the ES-2 cell line to 46.43%, the HEY-T30 cell line to 37.94%, and the OVCAR-3 cell line to 23.61%. This result is similar to the previously reported result where the cisplatin-resistant ovarian cancer cell line SKOV3 treated with 2.5 μM of MG-132 reduced the cell viability to 41.63% [15].

MG-132 induced apoptosis in ovarian cancer cell lines as evidenced by the presence of cleaved caspase-3 and cleaved PARP. Previously, the effect of MG-132 has been evaluated in esophageal squamous cell carcinoma (ESCC), oral squamous cell carcinoma (OSCC), osteosarcoma, adenoid cystic carcinoma (ACC), and many more cancers [28,29,30,31]. In oral squamous cell carcinoma (OSCC), MG-132 was found to block the degradation of p53 by UPP and induced apoptosis by nuclear factor kappa-light-chain-enhancer of activated B cells (NF-κb) pathway downregulation [29]. In the osteosarcoma cell line, MG-132 induced the downregulation of NF-κb and the PI3K/AKT (phosphoinositide 3-kinases/protein kinase B) pathway in addition to the upregulation of p21, causing the induction of apoptosis and the inhibition of the cell cycle [30]. On the other hand, the Nrf2/Keap1 (nuclear factor erythroid 2-related factor 2/Kelch-like ECH-associated protein 1) signaling pathway was found to be influenced in adenoid cystic carcinoma (ACC) by MG-132 [31]. In our study, we also found that MG-132 alone or in combination with ICA-1S can induce apoptosis in ovarian cancer cell lines. However, ICA-1S alone was unable to induce apoptosis significantly.

Mutations in p53 are common phenomena in ovarian cancer that can affect the treatment of it. Previous studies have reported that the mutation of the p53 gene can be correlated with the radioresistance of the ovarian cancer cell lines [32]. There are several experimental small molecules reported so far that influence upregulation of WTp53 or downregulation of Mutp53 [12,33,34]. Some of these drugs have shown unique modes of action, and some pathways are still unknown. Mutations in p53 can be structural mutants or contact surface mutants. The mutation in the ES-2 cell line (p53^S241F^) could be a contact mutation, and previously the ES-2 cell line was found to be moderately resistant to X-ray exposure [32]. In this experiment, ICA-1S was found to downregulate Mutp53 (45.79% of protein expression compared to the control) in the ES-2 cell line along with the downregulation of its target PKC-ι (71.3% expression of protein compared to the control). This phenomenon influenced our initial hypothesis that PKC-ι can phosphorylate p53 at the Ser-15 position, hence preventing the interaction of Mutp53 with MDM2 and subsequent proteasomal degradation, and administration of ICA-1S can inhibit this phosphorylation process. The phosphorylation of p53 at the Ser-15 position is important for the promoter activity of p53, DNA damage-induced activation of p53, and preventing the association of MDM2 in the normal cell line [35]. However, after applying the proteasomal inhibitor MG-132 to prevent this UPP-mediated degradation of p53 by ICA-1S, we found that MG-132 was further downregulating the Mutp53. A single dose (2 μM) of MG-132 lowered the expression of Mutp53 to 40.2%, and the combination of ICA-1S with MG-132 (2 μM) lowered the expression further to 19.14% (*p* < 0.001). In the HEY-T30 cell line with WTp53, ICA-1S also showed a pattern of downregulation of p53 (64.2%) and phospho-p53 (85%); however, it was to a lesser extent compared to the Mutp53. On the contrary, MG-132 significantly upregulated the WTp53 level (220.62%, *p* < 0.001) in the HEY-T30 cell line, even in the presence of ICA-1S treatment (157.60%). This phenomenon was also confirmed by immunofluorescence imaging of the p53 protein in the ES-2 and HEY-T30 cell lines. In the immunofluorescence, we can see Mutp53 level was prominent before the treatment in the ES-2 cell line. After the treatment with ICA-1S and MG-132, we observed a reduced fluorescence level of Mutp53. On the other hand, fluorescence level increased for WTp53 after the application of MG-132.

Previously, UPP was shown to stabilize the WTp53 in melanoma, indicating this is the major pathway of p53 turnover [36]. However, that study claimed MG-132 did not increase the level of phosphorylation of Ser-15 residue in p53, which is critical for the interaction of p53 with its ubiquitin ligase MDM2. In contrast, in our study both WTp53 and its Ser-15 phosphorylated (Pp53, 198.57%) form were elevated upon treatment with MG-132, showing WTp53 was mostly regulated via ubiquitination and subsequent proteasomal degradation in HEY-T30. This also indicated that the degradation of WTp53 by ICA-1S could be UPP mediated and related to the phosphorylation event at Ser-15, possibly mediated by PKC-ι. Co-immunoprecipitation with PKC-ι agarose-conjugated beads showed a strong association of WTp53 with PKC-ι in the HEY-T30 cell line, strongly suggesting PKC-ι might play a role in WTp53 phosphorylation (Figure 6C). Interestingly, the knockdown of PKC-ι by siRNA increased the protein level of WTp53 significantly, which emphasizes the role of PKC-ι as an oncogene in ovarian cancer. While this mechanism is not fully understood, it can be suggested that the significant downregulation of PKC-ι may upregulate some other tumor suppressor kinases that can phosphorylate WTp53 and stabilize it or may play a role in the transcription of the WTp53 (Figure 7).

The fact that MG-132 is inhibiting proteasomal degradation of WTp53 in the HEY-T30 cell line is supported by our result; however, the decreased level of Mutp53 cannot be explained via this pathway in the ES-2 cell line. In a previously published paper, it was shown that MG-132 can decrease the level of Mut53, and it was partially attributed to the increased level of HDM2 (MDM2 in humans) [16]. Another study previously suggested autophagy-mediated degradation of Mutp53 [37]. Hence, we aimed to explore this phenomenon in ovarian cancer cell lines. Beclin, phospho-ULK (Ser-757), and LC3A/B protein levels were determined via Western blotting to demonstrate the occurrence of an autophagy event. In the ES-2 cell line with Mutp53, the level of LC3 A/B II protein increased to 159% with the treatment of MG-132 and to 192.21% (*p* < 0.05) with the treatment of MG-132 and ICA-1S concurrently. Since the level of LC3 A/B II protein indicates an autophagic event and the level of LC3 A/B II production correlated with the Mutp53 degradation level, we suggest that in the ES-2 cell line, Mutp53 was degraded via the autophagic pathway by MG-132.

Even though ICA-1S reduced the level of P-ULK-1, no significant level of LC3 A/B II was detected to prove autophagy was occurring in these cell lines in the presence of ICA-1S alone. The pathway behind the degradation of Mutp53 by ICA-1S needs to be explored in future investigations.

PKC-ι is a known oncogene in many cancers; for example, in lung cancer, it is proved by the researchers that prevention of PKC-ι expression can block the transformed growth of lung cancer cell lines [38]. In triple-negative breast cancer, PKC-ι was found to play a critical role in invasiveness via the PKC-ι/RelA signaling pathway [39]. Overall, this study indicates that ICA-1S can inhibit PKC-ι and can degrade Mutp53; therefore, it can be a treatment option for both PKC-ι overexpressing and Mutp53-containing ovarian cancer. Since the combination of ICA-1S and MG-132 downregulated the Mutp53 and upregulated WTp53 while inducing apoptosis, this combination should be further explored for the therapeutic purpose of ovarian cancer.

## 4. Materials and Methods

### 4.1. Reagents, Antibodies, and Inhibitors

Primary antibodies were obtained as follows: PKC-ɩ (610175, BD Biosciences, Franklin Lakes, NJ, USA), PKC-ζ (sc-17781, Santa Cruz Biotechnology, Dallas, TX, USA), alpha tubulin (3873S, cell signaling, Danvers, MA, USA), beta actin peroxidase (A3854, MilliporeSigma, Burlington, MA, USA), p53 (2527S, cell signaling, Danvers, MA, USA), p-p53 Ser-15 (9284S, cell signaling, MA, USA), MDM2 (86934S, cell signaling, MA, USA), caspase 3 (9662, cell signaling, MA, USA), cleaved caspase 3 (9661, cell signaling, MA, USA), PARP (9542, cell signaling, MA, USA), cleaved PARP (9541, cell signaling, MA, USA), Phospho ULK-1 (6888S, cell signaling, MA, USA), Beclin (3495S, cell signaling, MA, USA), and LC3 A/B (12741 cell signaling, MA, USA). Anti-rabbit or anti-mouse antibodies conjugated with horseradish peroxidase (HRP) were used as secondary antibodies.

PKC-λ/ɩ-specific antibody (sc-17837, Santa Cruz Biotechnology, Dallas, TX, USA) conjugated to 25% agarose beads was obtained (Santa Cruz Biotechnology, TX, USA) for immunoprecipitation. p53 (sc-126) antibody conjugated with Alexa Fluor 488 was applied for immunofluorescence study.

Water-soluble tetrazolium salt-1 (WST-1) reagent was obtained from Roche (MilliporeSigma, St. Louis, MO, USA), and proteasome inhibitor MG-132 (Selleckchem, Houston, TX, USA) was used to inhibit UPP. For further use, MG-132 was diluted with sterile dimethyl sulfoxide (DMSO). ICA-1S was obtained from D.C. chemicals, Shanghai, China.

### 4.2. Ovarian Cancer Cell Lines

The ovarian cancer cell lines OVCAR-3, HEY-T30, and ES-2 were obtained from the American Type Culture Collection (ATCC; Manassas, VA, USA). OVCAR-3 was cultured with RPMI-1640 media (Catalog No. 30-2001, ATCC) supplemented with 20% fetal bovine serum (FBS, Corning, Glendale, AZ, USA) and insulin human, recombinant (ThermofisherScientific, Waltham, MA, USA). HEY-T30 was cultured with RPMI-1640 media (Gibco, Waltham, MA, USA), and the ES-2 cell line was cultured with McCoy’s 5a (ATCC, Manassas, VA, USA). RPMI-1640 and McCoy’s 5a media were supplemented with 10% fetal bovine serum (FBS). To prevent bacterial contamination, all media were supplemented with 1% penicillin (10,000 IU/mL) and streptomycin (10,000 μg/mL) antibiotics (Corning, AZ, USA). For experiments, cells were cultured on a T_75_ flask with 70 to 80% confluency and were incubated at 37 °C with 5% CO_2_.

### 4.3. Determination of Cell Viability by WST-1 and Cell Proliferation Assay

A WST-1 assay was performed to determine the cell viability in the presence of MG-132. Cells (5000 cells/well) were added to a Nucleon-Delta-treated 96-well plate (ThermoFisher Scientific, MA, USA). The next day, old media was replaced by 200 µL of new media, and cells were allowed to grow at 60–70% confluency. Cells were treated with MG-132 at a dose of 0.5 µM, 1.0 µM, 1.5 µM, and 2 µM and incubated for 18–24 h. After the treatment, old media with drugs were replaced with 90 µL of new media and 10 µL of WST-1 reagent. After adding the reagent, absorbance was taken at 480 nm at 1-h intervals for 4 h in a microplate reader (BioTek Synergy HTX Microplate Reader, Agilent Technologies, Santa Clara, CA, USA). The absorbance of the treated cells and control cells was compared and expressed as the percentage of cell viability.

To assess cell proliferation, cells were treated with varying doses of ICA-1S for 3 days for ES-2 and HEY-T30 cell lines and for 5 days for OVCAR-3. The treatments were conducted in T_25_ flasks. After the treatment period, the cells were collected, and their numbers were counted using the Nexcelom Cellometer Auto T4 Plus Cell Counter, Revvity, MA, USA. The dosage of the drug that reduced cell proliferation by 50% was identified as the IC_50_ (half-maximal inhibitory concentration) value.

### 4.4. Preparation of Cell Lysates and Western Blotting

Cells were seeded in 100 mm cell culture plates and were grown to 40–50% confluence before the treatment. Three days post-treatment with ICA-1S (IC_50_) and one day with MG-132 (0.5 or 2 µM), cell lysates were collected with lysis buffer (C7027, Invitrogen, Waltham, MA, USA) and processed. Bradford assay was performed using albumin standard and dye (Bio-Rad protein assay dye reagent concentrate) to quantify the protein concentration of each sample. The same concentrations of protein were loaded in gel wells, and SDS-PAGE (sodium dodecyl–sulfate polyacrylamide gel electrophoresis) was performed. Subsequently, the proteins were transferred to the 0.2 µm nitrocellulose membrane [40]. After adding the primary antibody and overnight incubation followed by the addition of a secondary antibody, the membrane was developed. The intensity of bands was determined by the Amersham Imager 600 software (GE Healthcare, Chalfont St Giles, UK) with auto exposure. The expression of each band was normalized with loading control of beta-actin peroxidase or alpha-tubulin developed on the same membrane.

### 4.5. Detection of p53 by Immunofluorescence

Cells were grown on 4-chamber slides (229164, CELLTREAT, Ayer, MA, USA) and were treated with ICA-1S (IC50), 2 μM MG-132, or ICA-1S (IC50) in combination with 2 μM MG-132. After 3 days of treatment with ICA-1S and one day of treatment with MG-132, cells were fixed with 4% paraformaldehyde and permeabilized with 0.1% Triton X-100. Alexa Fluor conjugated antibody was applied to cells with a 1:200 dilution for one hour. Following washing, slides were covered with ProLong™ Gold Antifade Mountant with DNA Stain DAPI (4′,6-diamidino-2-phenylindole, Invitrogen™), and coverslips were applied. Cells were then observed with a Nikon C2 confocal microscope at 60× magnification with oil immersion. All images were obtained through NIS-Elements imaging software version 5.21.00 (Nikon Instruments Inc., Melville, NY, USA) and were analyzed with ImageJ software 1.54 g [41]. To quantify the fluorescence level, the following formula was applied: final mean fluorescence intensity (MFI) = MFI of an ROI (region of interest)—MFI of Background [42].

### 4.6. Co-Immunoprecipitation

PKC-ɩ was immunoprecipitated from 1000 µg of protein samples of different cell lines with PKC-ɩ-specific antibody conjugated with agarose. After overnight gentle rocking with the conjugated beads, protein samples were centrifuged, and the first supernatant was kept to see the dissociated protein level. After a few washes, around 20 µL of pellet was prepared to see the association of proteins. PKC-ɩ beads without any protein sample were used as negative controls. Negative control, pellet, and supernatant were separated by SDS-PAGE and blotted to the nitrocellulose membrane. The association of protein p53 with PKC-ɩ was observed by adding a p53-specific primary antibody to the blot membrane.

### 4.7. Si-RNA-Based Knockdown of PKC-ι

ES-2 and HEY-T30 cells were seeded in 100 mm plates, and after growing to 60–70% confluency, cells were transfected with the transfection complex (siTran transfection reagent and 3 unique 27-mer siRNA duplexes specific for PKC-ɩ Locus ID 5584, SR321426, ORIGENE, Rockville, MD, USA). Trilencer-27 universal scrambled siRNA Duplex (SR30004, ORIGENE, MD, USA) was used as a negative control. After 5 h of treatment with the transfection complex, the media was removed and replaced with fresh media with FBS. After 48 h of growth, cells were obtained and processed for Western blotting as mentioned previously, and SDS-PAGE was performed.

### 4.8. Statistical Analysis

Statistical analysis was performed using GraphPad Prism 10.3.1. version. To compare the results of different treatments with the control, one-way ANOVA with Dunnett’s multiple comparisons was performed. To compare data of control and one type of treatment (i.e., for si-RNA), a t-test was performed. Mean ± SEM (standard error of means) of three independent trials were represented as bar graphs.

## 5. Conclusions

ICA-1S and MG-132 can reduce the proliferation and viability of ovarian cancer cell lines. MG-132 can degrade mutated p53 via autophagy while increasing the wild-type p53 by inhibiting proteasomal degradation. ICA-1S can degrade the Mutp53 as well, and knockdown of PKC-ɩ can increase WTp53 level. Together, ICA-1S and MG-132 should be further explored as a treatment option for ovarian cancer with mutated p53.

## Figures and Tables

**Figure 1 ijms-26-03035-f001:**
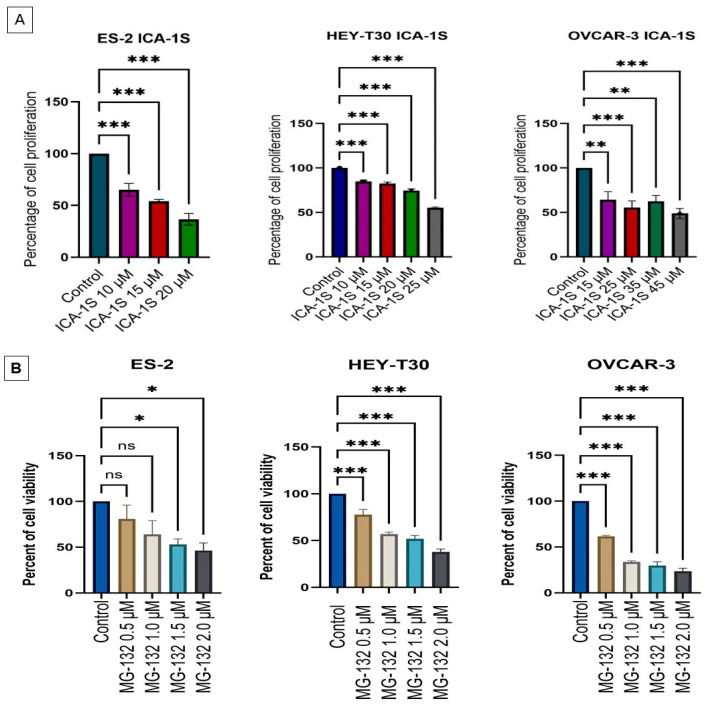
Cell proliferation assay with ICA-1S and cell viability assay (WST-1) with MG-132 of ES-2, HEY-T30, and OVCAR-3 cell lines. (**A**) ICA-1S has an inhibitory effect on the proliferation of ovarian cancer cell lines. One-way ANOVA following Dunnett’s multiple comparisons test showed that ICA-1S significantly reduced the number of cells after treatment compared to vehicle control. (**B**) 2 μM of MG-132 at a single dose significantly reduced the viability of treated groups compared to the control group. The bar graph represents the mean ± standard error of the mean (SEM) of at least three independent experiments with each cell line. ns, non-significant: * *p* < 0.05, ** *p* < 0.005, *** *p* < 0.001.

**Figure 2 ijms-26-03035-f002:**
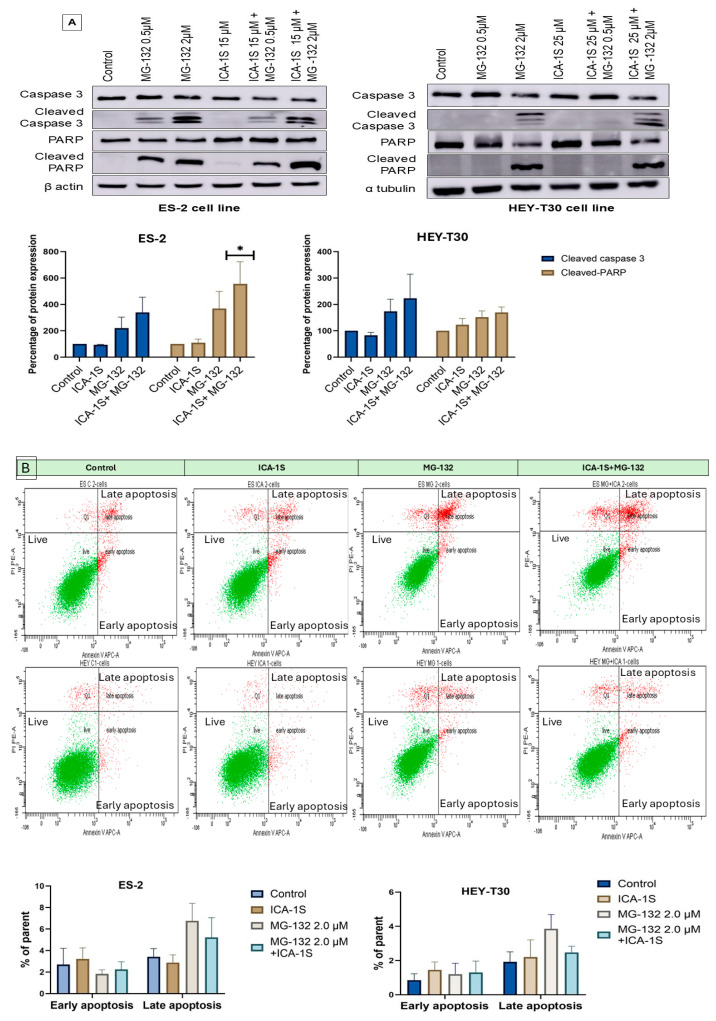
Treatment of MG132-induced cellular death in both ES-2 and HEY-T30 cell lines. (**A**) The combination of the ICA-1S (IC_50_) and MG-132 (2 μM) showed a high level of induction of apoptosis after treatment with drugs. Representative Western blot images of three independent experiments with a single drug or with a combination showed increased levels of cleaved caspase-3 and cleaved PARP. (**B**) Flow cytometry of Annexin V APC/PI analysis with a dot plot of individual cells showed mostly early apoptosis and late apoptosis with ICA-1S and MG-132 treatment, respectively. The bar graph represents the results of three independent trials as mean ± SEM, * *p* < 0.05.

**Figure 3 ijms-26-03035-f003:**
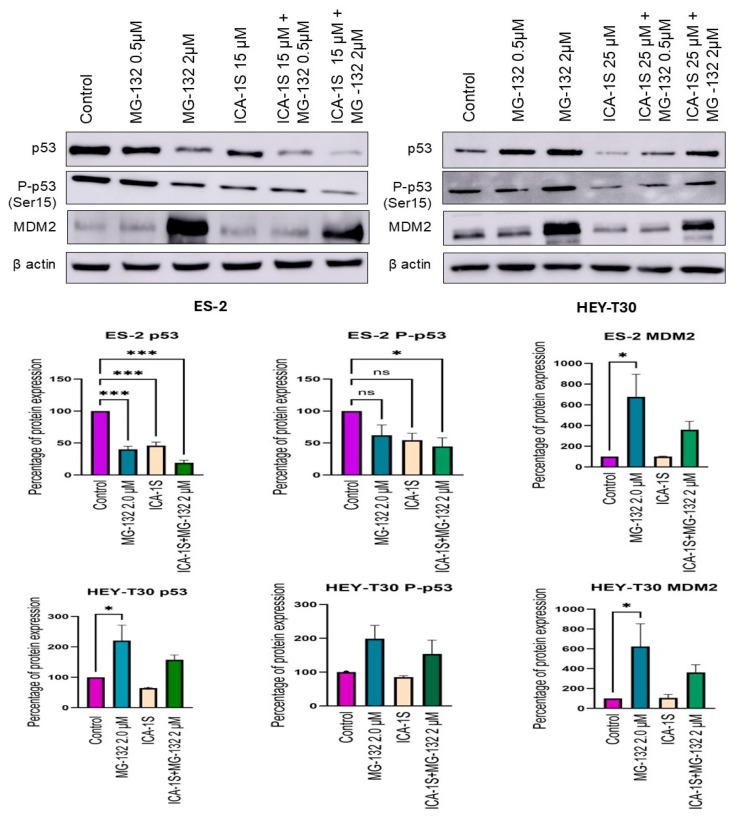
Mutp53 and WTp53 were regulated differently by ICA-1S and MG-132 in ES-2 and HEY-T30 cell lines. Western blots of p53, Pp53, and MDM2 of all treatments with control are presented. ICA-1S significantly reduced the Mutp53 level and reduced the WTp53 level to some extent. Mutp53 level was decreased with MG-132 treatment (ES-2) as well, whereas WTp53 level was increased with MG-132 treatment (HEY-T30). MDM2 levels were always upregulated with the MG-132 treatment. Bar graphs represent the mean ± SEM of the result of three independent trials after densitometric analysis, and statistical significance is presented as ns: non-significant, * *p* < 0.05, *** *p* < 0.001.

**Figure 4 ijms-26-03035-f004:**
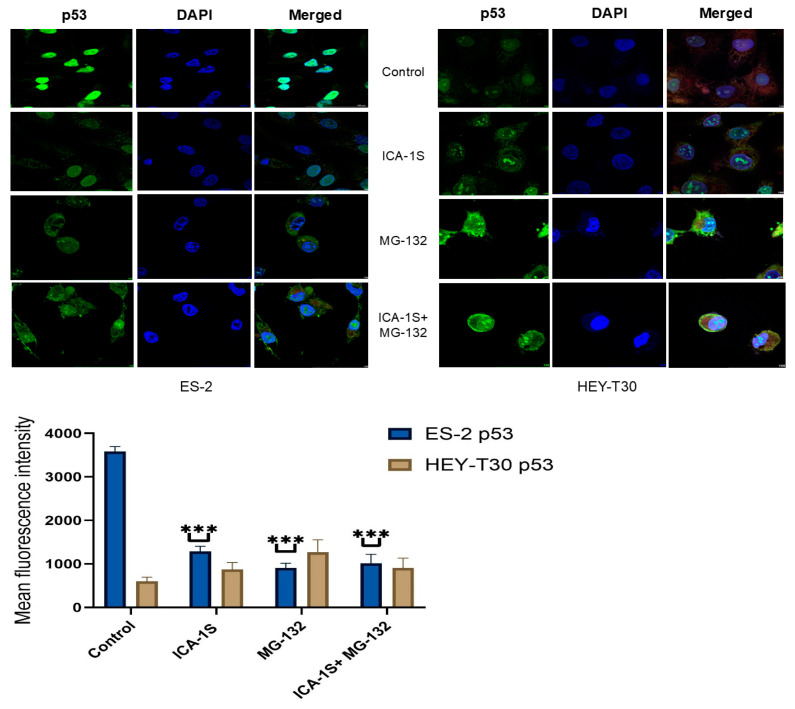
Immunofluorescence staining and visualization of p53 protein with DAPI staining of the nucleus in ES-2 and HEY-T30 cell lines. Graphical representation of the mean fluorescence intensity of three independent trials after analysis with ImageJ version 1.54 g shows the treatment of ICA-1S and MG-132 can significantly reduce the immunofluorescence level of Mutp53 in the ES-2 cell line. MG-132 increases the level of WTp53 in the HEY-T30 cell line compared to the control, *** *p* < 0.001.

**Figure 5 ijms-26-03035-f005:**
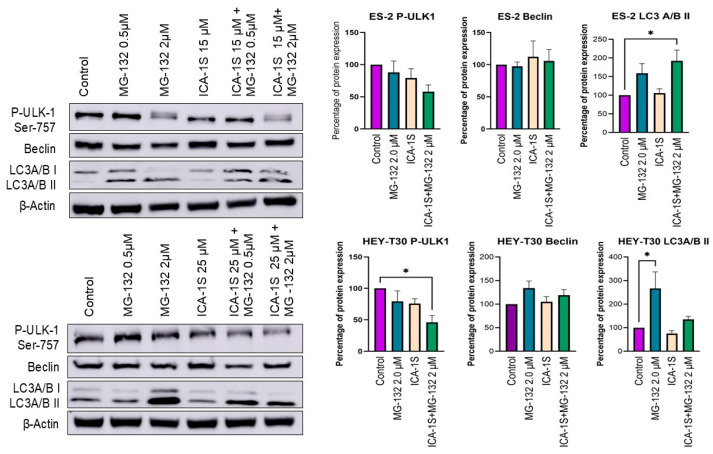
Level of autophagic pathway proteins P-ULK-1, Beclin, and LC3A/B in ES-2 and HEY-T30 cell lines after treatment with ICA-1S, MG-132, or the combination. The treatment of MG-132 induced the production of LC3A/B type II in both cell lines, indicating the occurrence of autophagy. Representative Western blots and bar diagrams are shown as mean ± SEM; * *p* < 0.05 significant level compared to the control.

**Figure 6 ijms-26-03035-f006:**
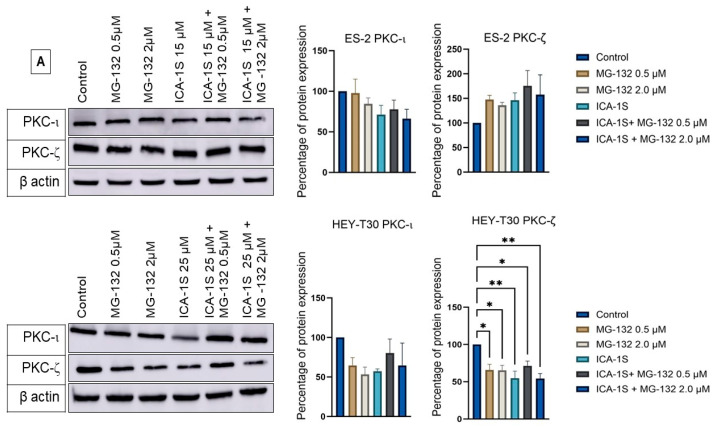
The level of atypical protein kinases in ES-2 and HEY-T30 cell lines and its relation with p53 protein. (**A**) Treatment with ICA-1S and with MG-132 lowered the levels of PKC-ι in both cell lines, whereas a different result was found for PKC-ζ in ES-2 and HEY-T30 cell lines. The level of PKC-ζ was significantly downregulated in the HEY-T30 cell line with ICA-1S and MG-132 treatment. (**B**) Knockdown of PKC-ι with specific siRNA reduced the level of PKC-ι. The level of WTp53 significantly increased with the downregulation of PKC-ι, whereas the Mutp53 level remained unchanged. (**C**) PKC-ι was strongly associated with p53 in the HEY-T30 cell line as evident by co-immunoprecipitation with PKC-ι-specific beads. Representative Western blots are presented. Bar diagrams of three independent trials are presented with mean ± SEM. * *p* < 0.05, ** *p* < 0.005.

**Figure 7 ijms-26-03035-f007:**
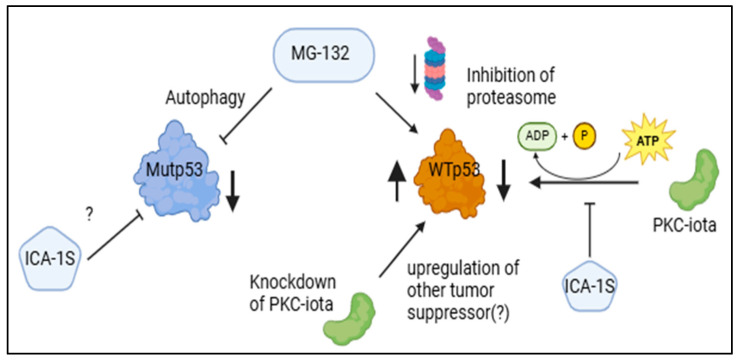
Schematic representation of the possible pathway of Mutp53 degradation. Proteasome inhibitor MG-132 degrades Mutp53 possibly by activating autophagy, whereas the degradation pathway of Mutp53 by ICA-1S remains elusive. WTp53 is upregulated by MG-132 via proteasomal inhibition, whereas WTp53 is downregulated by ICA-1S via blocking of phosphorylation. Knockdown of PKC-ι can upregulate WTp53 possibly via activation of other tumor suppressor kinases that phosphorylate WTp53 or by transcriptional upregulation.

## Data Availability

Data are contained within the article.

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
