# Peer review of "Proteasome Inhibitor MG-132 and PKC-ι-Specific Inhibitor ICA-1S Degrade Mutant p53 and Induce Apoptosis in Ovarian Cancer Cell Lines"

_ijms, 2025, doi:10.3390/ijms26073035_

Round 1
Reviewer 1 Report
Comments and Suggestions for Authors
This study investigates the mechanisms of action of ICA-1S and MG-132 in ovarian cancer cells. The research employs a comprehensive set of experimental methods, including cell proliferation assays, WST-1 viability assays, Western blotting, immunofluorescence, flow cytometry, RNA interference (siRNA), and co-immunoprecipitation (co-IP), providing robust evidence to support the findings. However, certain aspects require refinement.
Therefore, before accepting the manuscript for publication, I recommend a Minor Revision.
Regarding ICA-1S, does its effect on the p53 pathway resemble that of MG-132? Does it influence MDM2-p53 interactions? Beyond PKC-ι-mediated p53 regulation, are there additional unknown mechanisms involved? Furthermore, how does ICA-1S interact with existing chemotherapeutic agents, such as platinum-based drugs? A deeper exploration of these aspects would enhance the study's mechanistic insights.
Additionally, the Introduction should include a discussion of the role of PKC-ι in other cancers, such as lung and breast cancer, to emphasize its broader therapeutic relevance as a potential drug target.
Finally, some figures lack detailed statistical annotations. For instance, Figures 3 and 4 should include p-value indicators (e.g., "ns," "p < 0.05," "p < 0.01," "p < 0.001") to improve the clarity and rigor of statistical comparisons.
Author Response
Regarding ICA-1S, does its effect on the p53 pathway resemble that of MG-132? Does it influence MDM2-p53 interactions?
Answer: Thank you for the comment. As we can see in the figure 3 of manuscript, the level of p53 protein is different upon treatment with MG-132 and ICA-1S for the WTp53 in HEY-T30 cell lines. It is clear here that the pathway affecting WTp53 is different for ICA-1S and MG-132.
However, since MG-132 and ICA-1S both are downregulating the Mutant p53, we need more investigation on their mode of action. Our primary assumption is that MG-132 may induce autophagy dependent degradation of mutant p53 based on figure 5.
The expression of MDM2 is the highest when MG-132 is applied and the lowest when ICA-1S is applied. It is unlikely that MG-132 and ICA-1S has the same influence on MDM2. Previous report indicates that MDM2 undergo self ubiquitylation and degraded (Fang et al, 2000). Therefore, our data indicates that proteasomal degradation of MDM2 is inhibited with the application of MG-132. It is, however, not clear if ICA-1S can influence MDM2-p53 interaction. Co-IP with the treated samples may shed some light on this event. Thank you for the direction.
Fang, S., J. P. Jensen, R. L. Ludwig, K. H. Vousden, and A. M. Weissman. 2000. Mdm2 is a RING finger-dependent ubiquitin protein ligase for itself and p53. J. Biol. Chem. 275:8945-8951.
Beyond PKC-ι-mediated p53 regulation, are there additional unknown mechanisms involved?
Answer: We thank the reviewer for the question. It is highly possible that beyond PKC-i mediated p53 regulation, there are additional mechanisms since previous reports indicate several pathways are involved in p53 regulation. Our report is the first one to indicate that PKC-i can also play a role in p53 regulation.
Furthermore, how does ICA-1S interact with existing chemotherapeutic agents, such as platinum-based drugs? A deeper exploration of these aspects would enhance the study's mechanistic insights.
Answer: We thank the reviewer for the comment. It would be very interesting to look at the combination of ICA-1S with chemotherapeutic drugs such as taxane or cisplatin. In near future we will check this combination. For this study, it is beyond the timeframe to include this now.
Additionally, the Introduction should include a discussion of the role of PKC-ι in other cancers, such as lung and breast cancer, to emphasize its broader therapeutic relevance as a potential drug target.
Answer: Thank you for the suggestion. We have added lines 57-59 in the introduction and 316-319 in the discussion section.
Finally, some figures lack detailed statistical annotations. For instance, Figures 3 and 4 should include p-value indicators (e.g., "ns," "p < 0.05," "p < 0.01," "p < 0.001") to improve the clarity and rigor of statistical comparisons.
Answer: Thank you for the comment. In figure 3 “ns: non significant” has been added. Other p-value indicators have already been added.
Reviewer 2 Report
Comments and Suggestions for Authors
Mahfuza et al. demonstrate that PKC-i inhibitor ICA-1S and proteasome inhibitor MG-132 significantly reduce ovarian cancer cell proliferation through distinct molecular mechanisms. The study shows ICA-1S effectively decreases oncogenic PKC-i levels, while both compounds reduce mutant p53 in ES-2 cells. Notably, MG-132 increases wild-type p53 in HEY-T30 cells by inhibiting proteasomal degradation. These findings suggest that ICA-1S alone or in combination with MG-132 represents a promising therapeutic strategy for ovarian cancers characterized by mutant p53 and PKC-i overexpression. This work makes a valuable contribution to the field, worthy of publication in Int. J. Mol. Sci.
The following are some comments and suggestions that are given to improve the manuscript:
Comment 1: Has the research explored potential clinical applications of the ICA-1S and MG-132 combination? The paper suggests this combination could be promising for treating ovarian cancers with mutated p53 and overexpressed PKC-ι, but what would be the next steps toward clinical translation.
Comment 2: Since PKC-i knockdown increased wild-type p53 levels, would a higher dose of ICA-1S potentially overcome its effect of reducing wild-type p53.
Comment 3: The study utilized ES-2, HEY-T30, and OVCAR-3 cell lines, which may represent different types of ovarian cancer. It is worth exploring whether these findings can be validated in patient-derived primary cultures.
Comment 4: Considering that ICA-1S has shown promising results in vitro, it is possible to conduct in vivo experiments. The design of these experiments should evaluate the efficacy of ICA-1S alone or in combination with MG-132.
Comment 5: In addition to its interaction with p53, PKC-i also regulates other important cellular pathways. The effect of ICA-1S on these pathways may also contribute to its anti-tumor activity.
Author Response
The following are some comments and suggestions that are given to improve the manuscript:
Comment 1: Has the research explored potential clinical applications of the ICA-1S and MG-132 combination? The paper suggests this combination could be promising for treating ovarian cancers with mutated p53 and overexpressed PKC-ι, but what would be the next steps toward clinical translation.
Answer: We thank the reviewer for the valuable suggestion. Our next plan is to check the combination of ICA-1S with chemotherapeutic drugs such as platinum drugs. Our plan is also to investigate the combination of ICA-1S with MG-132. We are also interested to carry out an in vivo study with mice model.
Comment 2: Since PKC-i knockdown increased wild-type p53 levels, would a higher dose of ICA-1S potentially overcome its effect of reducing wild-type p53.
Answer: We appreciate the reviewer for this interesting observation. We are not sure what to expect with higher doses of ICA-1S on WTp53. Obviously, we can try to see the effect. However, there are reports that indicate that an inhibitor of a protein and a knockdown of a protein do not necessarily work in the same way (Lin, H.; 2023). So we may get different results with higher doses of ICA-1S as well.
Lin, H. (2023). Substrate-selective small-molecule modulators of enzymes: Mechanisms and opportunities. Current opinion in chemical biology, 72, 102231.
Comment 3: The study utilized ES-2, HEY-T30, and OVCAR-3 cell lines, which may represent different types of ovarian cancer. It is worth exploring whether these findings can be validated in patient-derived primary cultures.
Answer: We thank the reviewer for the comment. The ES-2 cell line is a clear cell carcinoma cell line, HEY-T30 is a taxane-resistant Papillary Cystadenocarcinoma and OVCAR-3 represents Adenocarcinoma. All of these cell lines were obtained from ATCC. Currently, our lab does not work with patient-derived primary cultures. However, we can share ICA-1S with research groups who have the facilities. Thank you for the suggestion.
Comment 4: Considering that ICA-1S has shown promising results in vitro, it is possible to conduct in vivo experiments. The design of these experiments should evaluate the efficacy of ICA-1S alone or in combination with MG-132.
Answer: Thank you for the comment. Sure, it is the next plan of the research group to conduct in vivo experiments with ICA-1S in mice model. Previously in vivo experiment with ICA-1S was published by this research group on glioblastoma with combination with Temozolomide (Dey, 2021).
Dey, A., Islam, S. A., Patel, R., & Acevedo-Duncan, M. (2021). The interruption of atypical PKC signaling and Temozolomide combination therapy against glioblastoma. Cellular Signalling, 77, 109819.
Comment 5: In addition to its interaction with p53, PKC-i also regulates other important cellular pathways. The effect of ICA-1S on these pathways may also contribute to its anti-tumor activity.
Answer: Thank you for the comment. Indeed PKC-i contributes to several tumorigenic pathways. Previously PKC-i was found to play role in glycolysis, hippo pathway, cyclin E expression, etc in ovarian cancer. This is quite possible that ICA-1S is also influencing these pathways. However, to explore other pathways more experiments are needed which is beyond the capacity of this research. In the future, we can look at those pathways with ICA-1S.